# Clinical, Virological and Immunological Responses after Experimental Infection with African Horse Sickness Virus Serotype 9 in Immunologically Naïve and Vaccinated Horses

**DOI:** 10.3390/v14071545

**Published:** 2022-07-15

**Authors:** Manuel Durán-Ferrer, Rubén Villalba, Paloma Fernández-Pacheco, Cristina Tena-Tomás, Miguel-Ángel Jiménez-Clavero, José-Antonio Bouzada, María-José Ruano, Jovita Fernández-Pinero, Marisa Arias, Javier Castillo-Olivares, Montserrat Agüero

**Affiliations:** 1Laboratorio Central de Veterinaria (LCV), Ministry of Agriculture, Fisheries and Food, Ctra. M-106, pk 1,4, 28110 Algete, Spain; mduran@mapa.es (M.D.-F.); rvillalba@mapa.es (R.V.); jbouzada@mapa.es (J.-A.B.); mruanor@mapa.es (M.-J.R.); 2Centro de Investigación en Sanidad Animal (CISA), Instituto Nacional de Investigación y Tecnología Agraria y Alimentaria (INIA-CSIC), Ctra. M-106, pk 8,1, 28130 Valdeolmos, Spain; pacheco@inia.es (P.F.-P.); majimenez@inia.es (M.-Á.J.-C.); fpinero@inia.es (J.F.-P.); arias@inia.es (M.A.); 3Tecnologías y Servicios Agrarios, S.A, (TRAGSATEC), 28006 Madrid, Spain; at_algete9@mapa.es; 4CIBER of Epidemiology and Public Health (CIBERESP), 28029 Madrid, Spain; 5Laboratory of Viral Zoonotics, Department of Veterinary Medicine, University of Cambridge, Cambridge CB3 0ES, UK; fjc37@cam.ac.uk

**Keywords:** African horse sickness, experimental infection, live attenuated vaccine, sero-neutralization test, ELISA, PCR, test precocity, virus isolation, tests performance characteristics

## Abstract

This study described the clinical, virological, and serological responses of immunologically naïve and vaccinated horses to African horse sickness virus (AHSV) serotype 9. Naïve horses developed a clinical picture resembling the cardiac form of African horse sickness. This was characterized by inappetence, reduced activity, and hyperthermia leading to lethargy and immobility–recumbency by days 9–10 post-infection, an end-point criteria for euthanasia. After challenge, unvaccinated horses were viremic from days 3 or 4 post-infection till euthanasia, as detected by serogroup-specific (GS) real time RT-PCR (rRT-PCR) and virus isolation. Virus isolation, antigen ELISA, and GS-rRT-PCR also demonstrated high sensitivity in the post-mortem detection of the pathogen. After infection, serogroup-specific VP7 antibodies were undetectable by blocking ELISA (b-ELISA) in 2 out of 3 unvaccinated horses during the course of the disease (9–10 dpi). Vaccinated horses did not show significant side effects post-vaccination and were largely asymptomatic after the AHSV-9 challenge. VP7-specific antibodies could not be detected by the b-ELISA until day 21 and day 30 post-inoculation, respectively. Virus neutralizing antibody titres were low or even undetectable for specific serotypes in the vaccinated horses. Virus isolation and GS-rRT-PCR detected the presence of AHSV vaccine strains genomes and infectious vaccine virus after vaccination and challenge. This study established an experimental infection model of AHSV-9 in horses and characterized the main clinical, virological, and immunological parameters in both immunologically naïve and vaccinated horses using standardized bio-assays.

## 1. Introduction

African horse sickness (AHS) is an infectious, non-contagious viral disease affecting all species of *Equidae*. It is caused by AHSV, a virus from the genus *Orbivirus,* family *Reoviridae*. AHSV is transmitted by biting midges of the genus *Culicoides*. Nine different serotypes (1–9) of AHSV have been identified. Four clinical forms of the disease have been formally described: (a) horse sickness fever, which is a mild form characterized by a transient febrile reaction without any other clinical manifestation; (b) a sub-acute cardiac form, characterized by high body temperature, edema, especially in neck and head, and petechiae of mucosal surfaces; (c) a hyper-acute pulmonary form with a predominance of respiratory signs such as severe dyspnea, high fever, presence of nasal exudate in the nostrils, and rapid death; and (d) a mixed form, which combines features of the cardiac and pulmonary forms [1,2,3].

The control of AHS is based on a rapid diagnosis, the prevention of insect biting, sound husbandry practices, the regulation of movements of equids between free and infected zones [4], and the systematic vaccination of horses in endemic areas. Currently, the only licensed vaccine in the world is a polyvalent live-attenuated vaccine (LAV), manufactured by Onderstepoort Biological Products (South Africa): *African horse sickness vaccine for horses, mules and donkeys* (Reg. No. G 0116 (Act 36/1947) Namibia: NSR 0586), which has been used with acceptable success in endemic countries. However, some concerns exist regarding their use in disease-free countries, specifically the risk of reversion to virulence, genome segment re-assortment between field and vaccine strains, and their lack of differentiation of infected from vaccinated animals (DIVA) capacity [5,6,7]. As a result, remarkable efforts have been made in the last two decades to develop new vaccine candidates that overcome the limitations of LAV [5,8,9,10,11].

AHS is endemic in sub-Saharan Africa with sporadic outbreaks occurring outside this area [3,12]. Although not all *Culicoides* species are equally efficient as biological vectors of AHSV [13], the worldwide distribution of these arthropods indicates that many parts of the world can be potentially affected by this devastating disease, as it was recently reported in Thailand and Malaysia [14,15,16]. In this context, the safe international movement of equids must rely on laboratory diagnostic methods with proven efficacy for the early detection of infection in a single animal.

The prevention and control of AHS is a priority at the international level. AHS is a listed disease of the World Organization for Animal Health (OIE) for its economic and animal welfare impact, and one of the six animal diseases for which a country-specific disease-free status is recognized by the OIE [16]. Moreover, given this strategic relevance, the European Union (EU) has categorized AHS as one of the five priority listed diseases in the new animal health law [17]. 

The Laboratorio Central de Veterinaria, Algete in Madrid, Spain, is the European reference laboratory for AHS and an OIE reference laboratory [18,19]. The EU/OIE reference laboratory provides scientific and technical support to the EU national reference laboratories of EU member states and harmonizes laboratory diagnostic methods as a key element for a rapid response to disease alerts. Within this framework, the production and standardization of reference material, as well as the organization of proficient testing schemes, require the availability of characterized anti-sera and tissues from infected and convalescent animals that can provide samples from actual disease scenarios.

In this study, we described the results of two experimental AHSV infection studies performed in naïve and vaccinated horses to achieve three objectives: (a) to characterize the clinical, immunological, and virological features of AHSV-9 infection in equine species under controlled conditions; (b) to generate diagnostic reference material (antisera, tissues, and blood samples); and (c) to obtain diagnostic and bio-assay data corresponding to these reagents. The recommended serological, molecular, and virological tests for laboratory diagnosis of AHS [20] were used in this study in the context of a controlled experimental infection. This work aimed at standardizing AHS experimental infection methods in equine species to facilitate the comparative evaluation of vaccine efficacy data obtained in different laboratories around the world. 

## 2. Materials and Methods

### 2.1. Study Design

Two experimental infections were conducted in consecutive years (Experiments 1 and 2). The aim of Experiment 1 was to establish a standard AHSV experimental infection in equids under specific conditions related to the immunological status of the animals, breed, age, sex, virus strain, challenge dose, and inoculation route. The main serological, molecular, and virological parameters were monitored using standard laboratory methods and immuno-assays. In Experiment 2, we used the standardized challenge model implemented in Experiment 1 to characterize the virological, serological, and clinical features of AHSV infection in horses vaccinated with a live attenuated vaccine. 

Two naïve horses were used in Experiment 1, and one naïve and two vaccinated horses were used in Experiment 2 (Figure 1). All experimental infections were performed by intravenous inoculation of a 4 mL dose containing 10 ^6.5^ tissue culture infectious dose (50%) (TCID_50_). In Experiment 2, horses were vaccinated according to the protocol described below and challenged together with a naïve horse (in a different box) 13 days after completion of the vaccination protocol (day 34 post-vaccination with Comb2/day 13 post-vaccination with Comb1). 

Animals were clinically monitored daily, and blood samples taken at periodic intervals for laboratory analyses. A quantitative assessment of the clinical signs mentioned below was used to establish, as objectively as possible, the need for palliative and analgesic treatments or applying euthanasia. After necropsy, organs were sampled for virus isolation, viral RNA, and VP7 antigen detection. Parameters investigated, tests used, and sampling scheme are summarized in Table 1.

Animal experimental protocols were approved by the Ethical Review Committee at the INIA-CSIC and Comunidad de Madrid (Permit number: PROEX 056/18 and PROEX 157/19 for Experiments 1 and 2, respectively). Both experiments were conducted in biosafety level 3 (BSL-3) animal facilities at Centro de Investigación en Sanidad Animal (CISA). The design of the experiments complied with the animal welfare considerations described in a royal decree 53/2013 [21] for this species. The animals were acclimatized in boxes for a period of 7–10 days before the start of experiments. During this time, they were observed daily by an equine veterinary surgeon and monitored to establish baseline values of rectal temperature, cardiac frequency, and respiratory frequency.

### 2.2. Horses

Two serological and virological naïve male horses (#1n and #2n) of Pottoka breed, 18 and 17 months old, respectively, were used in Experiment 1. In Experiment 2, one serological and virological naïve (#3n) and two vaccinated (horse #4v and horse #5v) Pottoka male horse of 19, 19, and 20 months old, respectively, were used.

### 2.3. Challenge Inoculum

The virus strain used for both challenges was derived from a lung exudate of a foal that died from the disease (Pirbright Institute Orbivirus Reference Collection: AHSV serotype 9 KEN/2006/02). The pathogenic potential of the strain had been demonstrated in a previous study conducted with a second isolate from the same individual [5]. The virus stock was grown in LCV by two serial passages in BHK-21 and KC cell lines before preparing the inoculum.

The inoculum was prepared by inoculation with 3 mL of virus stock containing 10^6^ TCID_50_/mL of infective virus on a 100% confluent monolayer of KC cells in a 75 cm^2^ flask, harvesting the supernatant after 8 days incubated at 28 °C. The final titre, expressed as TCID_50_, was determined by a standard end-point dilution assay in Vero cells and calculated using the Reed and Muench estimation method [22]. In addition, the exclusive presence of AHSV-9 in the inoculum was confirmed by using serotype specific RT-PCRs (see below) against all 9 AHSV serotypes. The same inoculum, kept at −80 °C, was used in Experiment 1 and Experiment 2. The titre was rechecked the day of inoculation.

### 2.4. Vaccination

In Experiment 2, a commercial polyvalent live attenuated vaccine (LAV) was used. The vaccine was supplied in two separated vials: combination 1 (Comb1) which included AHSV serotypes 1, 3, and 4 (AHSV-1, -3, and -4) and combination 2 (Comb2), which contained serotypes 2, 6, 7, and 8 (AHSV-2, -6, -7, and -8). The vaccine did not include either AHSV-5 or AHSV-9, but cross-protection to AHSV-5 and AHSV-9 was demonstrated after vaccination with AHSV-8 and AHSV-6, respectively [23,24]. Two animals (#4v and #5v) were immunized by subcutaneous route following the manufacturer’s instructions, except in the order of combined administration. Comb2 was administered first (day 0), and Comb1 21 days later (day 21). 

### 2.5. Clinical Monitoring

Respiratory frequency, pulse, heart rate, and rectal temperature were monitored twice a day (morning and afternoon). In addition, the following clinical parameters were monitored with the same periodicity: general condition of the animals (anorexia/loss of appetite, depression, immobility–recumbency, neurological signs, and sweating), circulatory signs (state of the mucous membranes, petechiae–ecchymosis–hemorrhage, subcutaneous edema, and pulse disturbance), respiratory signs (evaluation of noise by mediated auscultation, cough, expectoration, dyspnea–polypnea, and nasal exudate), ocular signs (conjunctivitis and subcutaneous edema, specifically in the palpebral, conjunctival, and supraorbital areas), and gastrointestinal signs (colic and diarrhea). In Experiment 2, rectal temperature as well as cardiac and respiratory frequencies were routinely recorded in the morning only due to personnel limitations.

### 2.6. Sampling

Peripheral blood samples were taken by vacuum tubes, with or without EDTA, at the intervals described in Table 1. EDTA blood samples were stored in cooling temperatures (2 °C–8 °C) until laboratory analysis, and whole blood samples were allowed to clot at room temperature and centrifuged at 2400× *g* for 5–10 min to obtain serum that was also stored at cooling temperatures (2 °C–8 °C) until testing.

Tissue samples (heart, lung, mediastinal lymph node, liver, spleen, kidney, and mesenteric lymph node) were taken after necropsy and stored at cooling temperatures. 

### 2.7. Virus Detection

#### 2.7.1. Serogroup-Specific Reverse-Transcription Real-Time PCR (GS-rRT-PCR)

The method published by Agüero et al. (2008) [25] and described in the European Union Reference Laboratory (EURL) procedure [26] was used for virus genome detection in peripheral EDTA blood, serum samples, and organ homogenates. This method employs two primers and a minor groove binder (MGB) Taqman probe targeted at the VP7 gene. Samples were classified as positive when a typical amplification curve was obtained and the cycle threshold (Ct) value was lower or equal to Ct value of 35 within 40 PCR cycles (Ct ≤ 35), inconclusive when 35 < Ct ≤ 40, and negative when no Ct was obtained. This method was validated according to OIE standards [27,28] and was one of the recommended methods by the OIE manual [20]. 

#### 2.7.2. Serotype-Specific Reverse-Transcription Real-Time PCR (TS-rRT-PCR)

TS-rRT-PCR methods were conducted for the nine AHSV serotypes in GS-rRT-PCR positive or inconclusive peripheral blood and organ homogenates samples. The methods used serotype-specific primers and probes targeting VP2 AHSV genes (Table 2) with the following conditions: 10 min at 48 °C and 10 min at 95 °C, followed by 40 cycles of 2 secs at 97 °C and 30 secs at 55 °C [39]. Samples were classified as AHSV positive, inconclusive, and negative according to the criteria described above.

#### 2.7.3. Virus Isolation

Serum samples with lower Ct values were selected to be inoculated in cell culture, without any treatment of the sera.

EDTA blood samples were washed and lysed by osmotic shock, as follows. After centrifugation of EDTA blood samples 1000 g/10 min/4 °C, the supernatant was removed, and the cell pellet washed three times with phosphate-buffered saline (PBS). Then the cells were lysed with 1 mL of sterile distilled water after maintaining the vial in ice for 10 min. Finally, after centrifugation at 12,000× *g* for 5 min at 4 °C and supernatant removed, 0.5 mL of PBS were added to the cell debris pellet, and the sample was stored at 2 °C–8 °C until testing [20,29]. Removing serum neutralizing antibodies by this procedure enhanced the recovery of the infectious virus present in the sample, which remained adsorbed to cell debris [30,31]. 

Organ tissues were processed to remove excess fat, connective tissue, and muscle; cut into pieces of the size of a lentil; and ground with glass beads and PBS (1 mL) using an automatic homogenizer. After centrifugation at 1000× *g* for 15 min at 4 °C, supernatants were passed through a 0.45 μm pore filter, and 1% of antibiotic-antimycotic solution 100 × (stabilized with 10,000 units penicillin, 10 mg streptomycin, and 25 μg amphotericin B per mL, sterile-filtered) was added and incubated for 20 min at room temperature. Finally, the organ homogenates were stored at +4 °C until testing [20,29].

Virus isolation was performed according to a standardized protocol [20,32]. Briefly, 0.2 mL of serum, washed and lysed EDTA blood 1:3 diluted in Minimum Essential Medium Eagle (EMEM) or organ homogenates diluted 1:9 in EMEM, were inoculated on 24-well tissue culture plates containing monolayers of 90% confluent Vero cells prepared the day before. After 1 h of incubation at 37 °C and 5% CO_2_, 0.8 mL of EMEM enriched with 1% glutamine, 1% non-essential amino acid solution 100×, 1% antibiotic-antimycotic solution 100× (stabilized with 10,000 units penicillin, 10 mg streptomycin and 25 μg amphotericin B per mL, sterile-filtered) and 2% fetal bovine serum heat inactivated (56 °C, 1 h) was added and incubated in the same conditions. Inoculated cell cultures showing no signs of cytopathic effect (CPE) by day 7 were passaged blind three times before considering them as negative. Samples were positive for the presence of AHSV when cytopathic effect was evident, and Ct value comparison between consecutive passages was compatible with the presence of virus replication. Virus isolates were typed by TS-rRT-PCR, as described above.

#### 2.7.4. ELISA for Antigen Detection

EDTA peripheral blood samples (only in Experiment 1) and organs after necropsy were analyzed using a commercial Capture ELISA [33] targeted to the outer core virus protein VP7. The test was performed as indicated by the manufacturer’s instructions. Samples with a corrected optical density (OD) higher than 0.200 were considered as positive, samples with OD lower than 0.150 were considered as negative, and OD between both values were considered inconclusive.

### 2.8. Detection of Circulating AHSV-Specific Antibodies

#### 2.8.1. VP7-Blocking-ELISA Test (b-ELISA)

Group-specific antibodies in sera were assessed using a commercial blocking ELISA [34,35]. The test was performed as indicated by the manufacturer’s instructions, following EURL guidelines [27]. The blocking percentage was calculated as follows: BP = (OD Neg Control–OD Sample) × 100/OD Neg control–OD Pos Control. Samples showing BP values > 50% were considered positive; samples with BP < 45% were considered negative; and samples with values between 45% and 50% were inconclusive. This kit was recently validated up to stage 3 [28] of the OIE validation pathway [36] and was recommended by the OIE manual [20].

#### 2.8.2. Seroneutralization Test (SNT)

Virus-neutralizing antibodies were detected according to a sero-neutralization method [20] following EURL protocol [37]. The test used the nine reference serotypes of AHSV. Positive sera to b-ELISA were assayed in duplicate by a two-fold dilution procedure starting at dilution 1/5. Titres were expressed as the inverse of the highest dilution of sera, allowing complete neutralization of 100 TCID_50_ of AHSV. Final titre of serum samples was determined according to Spearman–Kärber method [38,39].

### 2.9. Statistical Analysis 

For statistical assessment, chronological series of data between horses were compared (rectal temperature) using the Wilcoxon signed rank test for non-parametric data [40]. Differences supported by a *p*-value < 0.05 were considered statistically significant. 

## 3. Results

### 3.1. Clinical Signs and Pathology

The disease was successfully reproduced in naïve horses inoculated with AHSV-9 with clinical signs consistent with the cardiac form of AHS, although minor differences were observed in terms of individual response. 

Horse #2n developed clinical signs as of day two post-infection (pi), consisting of a slight increase in rectal temperature and elevated heart and respiratory rates. From day 5 pi onwards, the clinical signs worsened in #2n and then were observed in #1n. Horse #3n showed clinical signs from day 7 pi onwards. The disease progressed rapidly in the three animals, showing mild-to-moderate hyperthermia, increased cardiac and respiratory frequencies, the presence of abnormal sounds (crackles and wheezes) on lung auscultation, and dyspnea. Mild-to-moderate edema of the supraorbital fossae and eyelids were observed in the final stage (day 8 pi) with the animals displaying general depression, anorexia, and immobility, which rapidly evolved to recumbency, reaching clinical end-point criteria for euthanasia. Horses #1n and #2n had very low temperatures before becoming terminally ill (#1n at 9pi and #2n at 10 pi) (Figure 2).

Hyperthermia, tachycardia, and tachypnea, as well as the rest of signs, were more evident during afternoon monitoring. Therefore, rectal temperature was significantly higher when comparing afternoon vs. morning values in the same individual (horse #1n-*p*-value = 0.0032; horse #2n-*p*-value = 0.0006) (Figure 2).

The differences in rectal temperature (morning check) were statistically higher in #3n, as compared to #1n (*p* value = 0.003), but not to #2n (Figure 2).

The pathological examination after necropsy revealed lesions compatible with the cardiac form of AHS: supraorbital edema, congestion of ocular conjunctiva, severe hydropericardium (#1n, #2n, #3n) with an abundant presence of transudate, hydrothorax, and ascites (#1n, #2n); congestion and edema of heart, lungs, liver, spleen, and kidney; the presence of hemorrhagic areas in the lungs, heart, and kidneys; and the presence of frothy fluid in the trachea (#1n, #2n, #3n) and in the pulmonary parenchyma (#2n, #3n).

The vaccinated horses (#4v, #5v) did not show abnormal clinical parameters after vaccination or after challenge, except horse #4v that showed a mild increase in respiratory frequency (32 breaths per minute) on day 1 after vaccination with combination-1 (pv-C1), and horse #5v with a weak increase in cardiac frequency (52 beats per minute) on day 8 post-vaccination with combination-2 (pv-C2). Locally, a painless mild swelling was observed at the inoculation site on days 1 and 2 after administration of each combination. 

The horses were euthanized at the scheduled time, and the necropsy revealed mild congestion of heart, liver, spleen, and kidney.

### 3.2. Laboratory Parameters 

#### 3.2.1. Virus RNA Detection in EDTA Blood

In unvaccinated horses (#1n, #2n and #3n), the GS-rRT-PCR consistently detected AHSV RNA in blood samples from day 4 pi (#1n, #2n) and 7 pi (#3n) onward with decreasing Ct values until the end of the study period. No significant differences (Ct values) were observed between the animals (Table 3).

In the vaccinated horses (#4v and #5v), virus nucleic acid was detected by GS-rRT-PCR before the challenge with no differences (Ct values) between horses. Horse #4v showed detectable RNAemia from day 13 pv-C2 and horse #5v from day 21 pv-C2. Notably, this post-vaccination RNAemia was detected in absence of hyperthermia. After challenge, virus nucleic acid continued being detectable in both horses until the end of the experimental infection with variable Ct values. Both pre- and post-challenge Ct values were above Ct 30, except in horse #5v on the day of euthanasia, which had a Ct of 29.6. (Table 4).

Regarding the TS-rRT-PCR, vaccine virus serotypes contained in Comb2 (AHSV-2, 6, 7-and 8) were detected as follows: serotype 8 was consistently detected in horse #4v from day 13 pv-C2 until day 30 pv-C2, and serotype 2 in horse #5v from day 24-to 27 pv-C2, and after challenge, in both horses until the end of experimental infection. Serotype 6 was detected from day 16 to 24 pv-C2 in #4v. Serotype 7 was never detected. Serotypes included in Comb1 (AHSV-1, 3 and 4) were never detected in vaccinated animals. Moreover, horse #5v was TS-rRT-PCR-positive to serotype 9 (serotype of the challenge strain) on the day of euthanasia (16 dpi), as shown in Table 4.

#### 3.2.2. Virus RNA Detection in Blood Sera

Although serum is not the sample of choice for orbivirus detection, AHSV RNA was detected from days 8 and 9 pi in all unvaccinated infected horses, although the Ct values were significantly higher than those observed in paired EDTA blood samples (Figure 3).

#### 3.2.3. Virus Isolation in EDTA Blood

The AHSV-9 challenge strain was isolated from blood as early as day 3 pi (#2n), 4 pi (#1n), or 7 pi (#3n), and until the day of euthanasia in unvaccinated horses (Table 3).

In the vaccinated horses, serotype 8 was isolated in the blood of horse #4v on day 16 pv-C2 and, notably, serotype 2 on day 37 pv-C2, three days after challenge (day 3 pi). Regarding horse #5v, the vaccine strains were not isolated from peripheral blood, but the AHSV-9 challenge strain was isolated from blood on day 16 pi (day of euthanasia) (Table 4).

#### 3.2.4. Virus Isolation in Blood Sera

Consistent with the GS rRT-PCR results, the virus isolation technique was performed on the two sera with the lowest Ct values (#1n at day 9 pi and #2n at day 10 pi), obtaining a positive result in the serum sample of #2n at day 10 pi.

#### 3.2.5. Virus Antigen Detection in EDTA Blood (Only in Experiment 1)

ELISA-Ag failed to detect the virus antigen in EDTA blood in Experiment 1 (horse #1n and #2n). Therefore, the test was not conducted in Experiment 2.

#### 3.2.6. Virus Detection and Virus Isolation in Organs after Necropsy

AHSV was detected in all organs of the unvaccinated horses by GS-rRT-PCR and antigen ELISA. Infective virus was recovered from the spleen and the lungs (#1n, #2n, and #3n), heart (#1n), the kidney (#2n and #3n), the liver (#2n and #3n), and the mediastinal and mesenteric lymph nodes (#2n and #3n) (Table 5). The virus isolated from tissues was confirmed to be AHSV-9 by TS-rRT-PCR.

In the vaccinated horses (#4v and #5v), ELISA-Ag was negative in the spleen (not performed for other organs). The GS-rRT-PCR detected the AHSV RNA in the spleen, the lung, the liver, the heart, and the mediastinal lymph nodes of both horses, and in the kidney and the mesenteric lymph nodes of horse #5v, although the Ct values were remarkably high. The TS-rRT-PCR detected the AHSV-9 RNA in all the organs of horse #5v and in the liver of horse #4v. AHSV-2 was detected in the spleen and the liver of horse #4v, and in the spleen as well as the mediastinal and mesenteric lymph nodes of horse #5v. The virus isolation from the organs of vaccinated horses was negative, except from the spleen of horse #5v where AHSV was isolated and confirmed as the serotype 9 by TS-rRT-PCR (Table 6).

#### 3.2.7. Antibody Detection in Sera

The naïve infected horses remained negative according to b-ELISA during the entire experiment, except horse #2n who showed a doubtful response on day 9 pi and a weak positive result on day 10 pi. 

In the vaccinated horses, VP-7-specific antibodies could be detected by b-ELISA in horse #4v from day 21 pv-C2, before the administration of Comb1. In contrast, horse #5v did not show VP-7-specific antibodies until it received the second vaccination, 9 days pv-C1 (day 30 pv-C2). A slight increase in b-ELISA signal (blocking percentage) was detected after challenge (Figure 4).

The positive/inconclusive sera to b-ELISA were tested by sero-neutralization test. Before challenge, only #4v showed low SN titres against serotype-6 (titre range: 5–15) and serotype 8 (titre range: 7.5–15). After challenge, horse #4v maintained SN titres against serotype 8 (15), slightly increased titres against serotype 6 (up to 20–30), and for the first time, titres against serotype 1 (7.5 day 14pi), serotype 2 (7.5 or 30 day 11 or 14 pi, respectively), serotype 4 (7.5–10 days 9–14 dpi), and serotype 5 (7.5–10 days 9–14 dpi). No antibodies were detectable against serotype 3, 7, and, remarkably, serotype 9. In horse #5v, only antibodies against serotype 2 (7.5–15) were detected from day 9 pi to 16 pi (Table 7).

## 4. Discussion

The experimental infections described for this study were successful in reproducing the disease in two consecutive experiments under controlled conditions and allowed us to obtain clinical and laboratory parameters after vaccination with LAV and/or a challenge with virulent AHSV-9.

Naïve horses developed the cardiac form of African horse sickness, consisting of an initial mild-to-moderate syndrome that became more severe after day 7 post-infection. Therefore, the horses became recumbent and lethargic by days 9 and 10 pi, and they had to be euthanized. Hyperthermia was the earliest clinical sign, followed by the increase in cardiac and respiratory rates, and the presence of mild-to-moderate edema in the eyelids and supraorbital fossae. Dyspnea, anorexia, and lethargy developed in the last days of the process. Although slight differences between horses were observed, the clinical and pathological pictures were very similar in all three animals in both experiments. This common pattern of disease in Experiment 1 and 2 pointed to a relation between virus strain and the clinical picture in the host. Moreover, similarities with other studies that used an AHSV serotype 9 strain isolated from the same infected animal [5] emphasized the importance of standardizing challenge materials and procedures (host: breed, sex, and age; virus: challenge strain, route, and dose), and that such standardization is feasible. This is very important for a disease such as AHS, which is very difficult to study experimentally in vivo (due to logistic, financial, ethical, and biosafety constraints). The standardization of experimental infections could enable a more reliable comparison of the challenge data. This would simplify vaccine efficacy testing and accelerate vaccine development.

In Experiment 1, we compared the clinical conditions of horses (#1n and #2n) between morning and afternoon checks and corroborated that rectal temperatures and clinical signs were more prominent in the afternoon. This was a useful consideration when preforming clinical examinations of potential AHS horses, at least in experimental infections, although in Experiment 2, unfortunately, afternoon checks could not be performed; therefore, only morning clinical data were compared among vaccinated and naïve horses after challenge.

Regarding vaccinated horses (#4v, #5v), the live attenuated vaccine did not induce remarkable side effects, and it protected animals from severe disease despite the low or undetectable levels of circulating VNAb at the time of the challenge. Even though serotypes 5 and 9 were not included in Onderstepoort LAV, previous studies [23] have demonstrated the solid protection conferred against serotypes 5 and 9 after vaccination with serotypes 8 and 6, respectively, using a booster protocol (re-vaccination with Comb1 and Comb2 at day 49 and 70).

Due to the short time period between the second dose of the vaccine and the challenge (13 days), we changed the order of the recommended vaccine protocol by administering the combination containing serotype 6 (Comb2) first to induce protection against AHSV-6 and, consequently, increased cross-protection efficacy against AHSV-9. Although our study showed that the cross-protection induced by the vaccine with the change in order (first Comb2 and, later, Comb1) prevented clinical disease after challenge with AHSV-9, the absence of information regarding protection results using the standard protocol (Comb1 followed by Comb2) and challenging the animals in less than two weeks after completing the vaccination protocol did not allow for any comparative conclusion to be reached.

The assessment of the vaccine efficacy was not among the objectives of this study due to the shortcoming of the experimental design, such as, only two vaccinated animals, early challenge (accordingly to vaccine manufacturer´s instruction, immunity starts to develop two-to-three weeks after complete inoculation, and protection against some of the virus types are achieved after four weeks) and the exclusion of a LAV vaccinated-only horse. However, the AHSV-9 challenge strain was still detected and isolated from the bloodstream of horse #5v on the day of euthanasia and its isolation in the spleen as well as RNA detection in most of the organs. This indicated that the vaccine-induced immunity did not completely protect against infection under these experimental conditions. Consistent with previous studies [5], viral RNA (GS- rRT-PCR) and the infectious virus were detected in the blood and various organs of vaccinated and challenged horses, despite these animals being clinically protected. 

Previous studies of orbiviral diseases indicated that, in general, the viral RNA load in serum was markedly lower than in EDTA blood samples [41], and it was confirmed by GS rRT-PCR in our study. However, we observed infectivity in a serum sample from one of the animals during the peak of RNAemia, which should be considered when assessing the bio-risk of processing serum samples from AHS-suspected horses, especially in laboratories located in disease-free countries.

In this study we detected vaccine viruses by rRT-PCR in biological samples from vaccinated horses. Moreover, in horse #4v, serotype 8 was isolated p and serotype 2 three days after the challenge, suggesting a high load of these serotypes in the bloodstream during those specific days. This extended RNAemia was also observed by Weyer et al. (2012) [42] as well as Weyer et al. (2017) [24] in vaccinated horses under breeding conditions with an equivalent vaccination protocol (Comb 2, day 0; Comb 1, day 28). As in our study, the lowest GS-rRT-PCR Ct values were reached in the third week post-vaccination, even before the administration of combination 1. Although in the study mentioned (Weyer, 2017) that horses were not challenged after vaccination, the follow-ups of the serotypes in the blood by TS-rRT-PCR from week 4 to week 16 were consistent with our findings. Serotypes 8 and 2 were detected, whilst serotypes 7, 1, and 4 were not. In contrast to our results, serotype 3 was detected in the very end of the study (week 16). Further studies are needed to elucidate whether viremic-vaccinated horses represent a potential risk of onward transmission in the field.

Moreover, serotypes 9 and 2 nucleic acids were simultaneously detected in the spleen and the lymph nodes of horse #5v, and in the liver of horse #4v. Overall, these findings indicated that the vaccine-derived and challenge viruses could co-circulate in the animal for a period of time, favoring the possibility of genome-segment reassortment and a reversion to virulence of the attenuated vaccine strains. Future studies using whole genome sequencing could improve our understanding of in vivo gene-segment re-assortment and its biological consequences. This mechanism of emergence of new virulent strains has been well documented through studies using this technology in AHS isolated in the context of recent outbreaks reported in South Africa [7].

Despite the artificial components inherent to experimental infections, such as the use of high-titre cell-culture-grown virus for reproducing an AHSV infection administered intravenously, this type of study provides useful models for assessing the performance of laboratory tests in terms of precocity and their capability to detect a pathogen in the course of a disease, in vaccinated and/or infected individuals. In addition, the immunity developed after vaccination can be accurately measured.

The demonstration of individual animal freedom from AHS infection is key for a safe international movement of horses, and therefore, an ideal diagnostic test must detect the viral pathogen and virus-specific antibodies early after infection and consistently thereafter. This ability was recognized in PCR, virus isolation, and serology methods [20]. Our study showed that in naïve, unvaccinated horses, both GS-rRT-PCR [25] and virus isolation were effective for the detection of AHSV in the bloodstream as early as 3 – 4 days post-infection. By contrast, serogroup-specific VP7 antibodies were undetectable by b-ELISA in 2 out of 3 horses over the course of the disease (9-to-10 days after which, the animals were euthanized due to the advanced state of illness), as expected. Only horse #2n started showing low levels of detectable antibodies the day before euthanasia (9 – 10 dpi). In the experimental infection conducted by Alberca et al. (2014) [5], non-vaccinated horses were euthanized at days 5 and 6 without showing b-ELISA antibodies to AHSV-9 but showing a response to GS-rRT-PCR. The same profile was reported by Van Rijn et al. (2018) [11], who did not find seroconversion to b-ELISA in two control animals that died at days 6 and 8 after experimental infection with AHSV-5. Other authors have reported earlier detections of VP7 antibodies (7 dpi) using an indirect anti-protein G-ELISA [43]. However, the low comparative diagnostic precocity of antibody ELISA tests, as compared to the GS-rRT-PCR, emphasizes the importance of combining serological tests with strict quarantine and GS-rRT-PCR testing to detect the early stages of the disease in animals moving from endemic territories to disease-free zones. Ideally, the combined use of GS-rRT-PCR and ELISA would offer the greatest guarantees in the control of animals prior to movement.

The laboratory confirmation of clinical cases is important to promptly recognize the presence of an AHS infection, especially in disease-free zones. If possible, more than one test should be performed to diagnose an outbreak of AHS, especially an index case [20]. In our study, both GS-rRT-PCR and virus isolation demonstrated optimal sensitivity to the early detection of infection in blood samples of live naïve horses (#1n, #2n, #3n). Similar results have been reported by other authors [5,23,44]. Regarding the detection of the pathogen in organs after necropsy, virus RNA and virus antigen were detected in all the organs of the three naïve infected horses by GS-rRT-PCR and antigen ELISA, while the virus could not be consistently isolated in all organs, although it was in the spleen and the lung. Undoubtedly, the combined use of GS-rRT-PCR and virus isolation would be ideal in the first characterization of disease outbreaks. However, for monitoring the evolution of the disease, the GS-rRT-PCR method would offer the advantages in terms of cost, rapid results, and sample processing capacity.

The ELISA for antigen detection failed in detecting AHSV in the bloodstream, which limited its usefulness in live horses with symptoms, but showed a good diagnostic sensitivity in organs. Although at the time of this study, the test was pending validation for all AHSV serotypes in clinical samples [33], it would be useful in those laboratories that do not implement PCR technology for confirmation of disease in dead animals.

As far as post-vaccination immune response was concerned, b-ELISA detected positive animals between 21 and 30 days after vaccination, although from that point on, the animals remained positive throughout the experiment. An aspect of the immune response unexplored in African horse sickness has been the ability of the ELISA test to detect a significant increase in antibody titres if an LAV-vaccinated animal becomes infected by a wild-type virus strain, thereby identifying potentially dangerous animals from an epidemiological point-of-view. 

Although only two vaccinated animals were employed, there was a remarkable individual variation in the detection of sero-neutralizing antibodies after vaccination, which agreed with previous studies [23,24]. In horse #5v, the response was undetectable during the vaccination stage and only detectable to serotype 2 at low titre, 8 days after challenge; in horse #4v, the neutralizing antibodies response was detectable to all serotypes contained in the vaccine, except serotypes 7 (Comb2) and 3 (Comb1). These sero-neutralizing antibody profiles were comparable to that reported by Weyer et al. (2017) [24], as previously discussed. In that study, there was a marked disparate response between twelve weanlings without a complete seroconversion of the group to any serotype. As in our study, antibodies against serotype 7 were undetectable. However, we obtained lower titres, as compared to that study, as well as to the study of Von Teichman et al. (2010) [23], which used, as an antigen of the test, the original isolates for the development of the vaccine seed material. It was suggested that sero-neutralizing antibody titres would correlate with vaccine protection, so that a titre of 16 or higher would be protective [45]. Indeed, neutralizing antibody titres would be a very useful parameter to evaluate the in vitro efficacy of the different candidate vaccines, despite the intrinsic complexity of SNT, but the variability of titres observed in different studies have suggested the need for the improved standardization of test procedures for comparability (e.g., virus strain collection used as antigen, cell lines, volumes of reaction).

When we compared the results of serotype detection between SNT and TS-rRT-PCR, we found a basic agreement. The most detected serotypes in TS-rRT-PCR in horse #5v (S-2) and in horse #4v (S-2, S-6 and S-8) corresponded to the serotypes producing higher titres by SNT. 

Interestingly, horse #5v did not show neutralizing antibodies to serotype 6, even though the animal appeared to be clinically protected against the cross-reactive AHSV serotype 9 challenge strain. This absence of detectable neutralizing antibodies in animals showing protection to the challenge virus was also found by Von Teichman et al. (2010) [23] using the standard vaccination protocol (first shot, combination 1, and second, combination 2). This underlined the role of a cell-mediated immune mechanism in the protection against infection in vaccinated individuals.

Although it appeared that the vaccinated horses were clinically protected in this experimental infection, both the challenge and the time of euthanasia of these horses were too early to be confident that the adaptive immune response had been fully established. Additional studies with longer time periods from vaccination to challenge as well as longer post-challenge durations are required to verify that the horses would survive.

This study represented a further step towards the necessary standardization of experimental infection models in a disease as relevant as African horse sickness as well as for the evaluation of the performance of diagnostic methods in infected and vaccinated animals. In addition, the tissue samples obtained provided reference materials that will be essential as we move towards the standardization of laboratory testing.

## Figures and Tables

**Figure 1 viruses-14-01545-f001:**
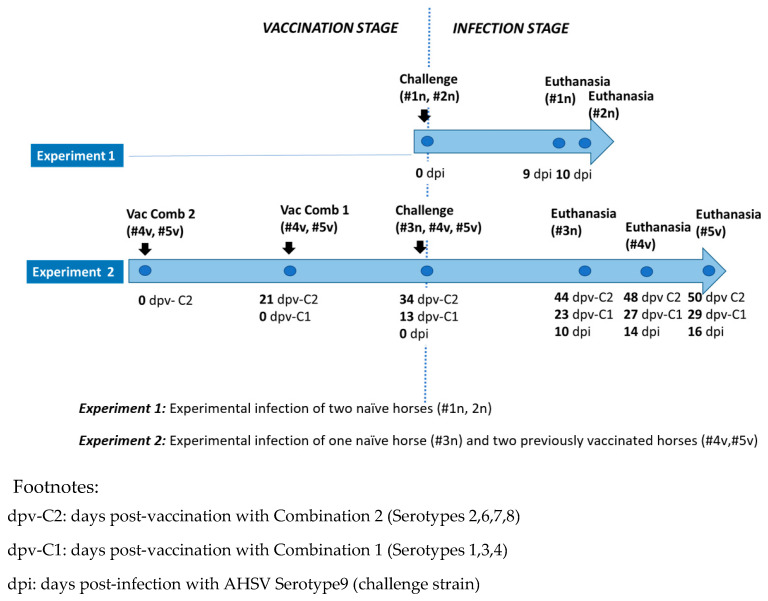
Study design and milestones.

**Figure 2 viruses-14-01545-f002:**
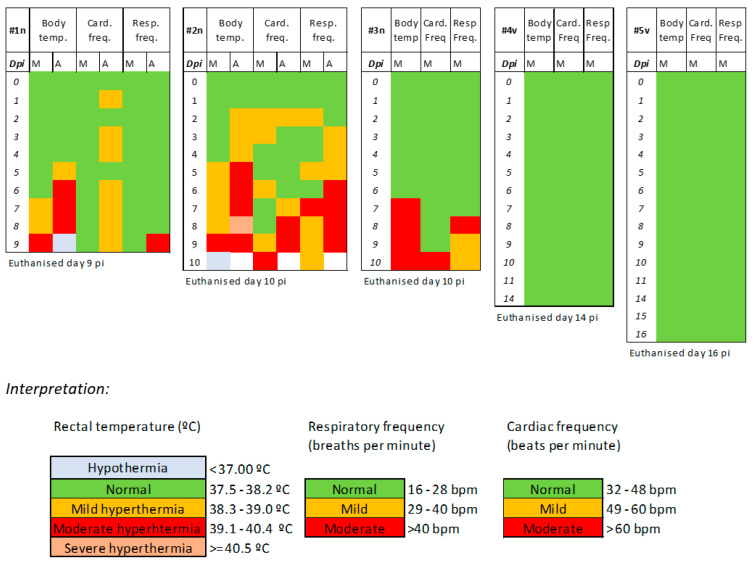
Clinical monitoring of horses (Infection stage).

**Figure 3 viruses-14-01545-f003:**
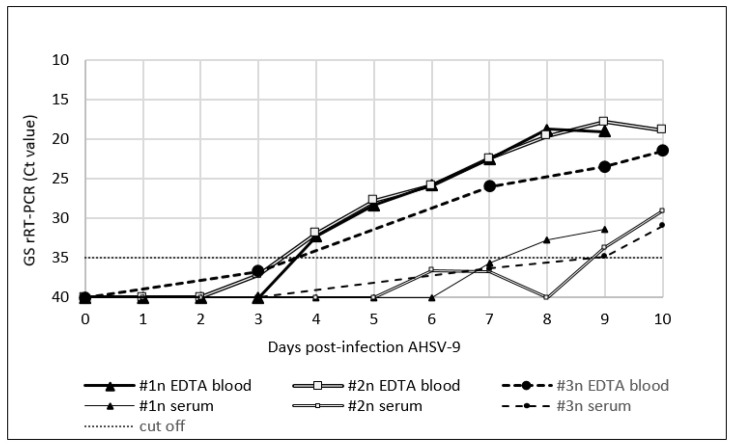
Comparation of RNAemia (GS rRT-PCR) in EDTA blood and serum samples in naïve horses experimental infected (#1n, #2n and #3n).

**Figure 4 viruses-14-01545-f004:**
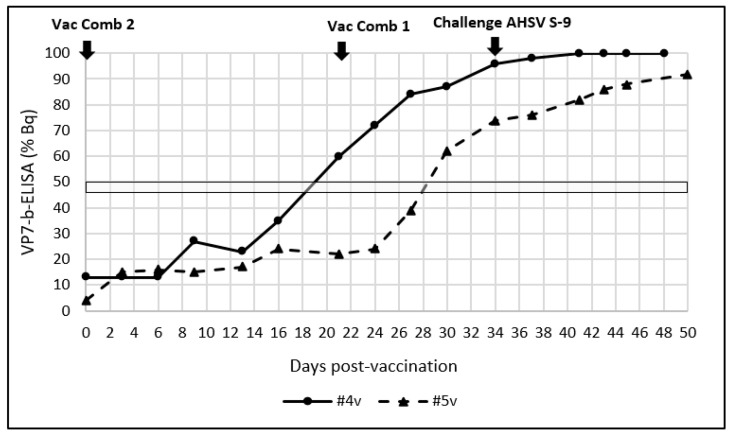
Detection of antibodies by VP7 Blocking-ELISA in vaccinated horses #4v, #5v (Vaccination and infection stages).

**Table 1 viruses-14-01545-t001:** Parameters investigated, tests, and sampling scheme in Experiments 1 and 2.

Parameter (Sample)	Test ^†^	Sampling Scheme
Detection of virus in peripheral blood (EDTA blood sample) (Serum sample) *	GS-rRT-PCR TS-rRT-PCR (only vaccinated horses) Virus isolation in cell culture ELISA for Antigen detection (only in Experiment 1)	Horses #1n. #2n: 0.1.2.3. 4. 5.6.7.8.9.10 dpi Horse #3n:0.3.7.9.10 dpi Horses #4v #5v: 0.3.6.9. 13.16.21.24.27.30.34.37.41.43.44.45.48.50 dpv-C2
Antibody detection in sera of peripheral blood (Serum sample)	VP7-Blocking-ELISA (b-ELISA) Seroneutralization test (SN)	
Post-mortem detection of virus in organs (Tissue homogenate sample)	GS-rRT-PCR TS-rRT-PCR (only vaccinated horses) Virus isolation in cell culture ELISA for Antigen detection	Euthanasia day: Horse #1n: 9 dpi; Horse #2n: 10 dpi; Horse #3: 10 dpi; Horse #4v: 14 dpi; Horse #5v: 16 dpi

(^†^) All tests except ELISA for antigen detection are accredited according to ISO 17.025. (*) Although serum is not the choice sample for orbivirus detection, GS rRT-PCR was conducted on horses of Experiment 1 to evaluate the presence of AHSV in that fraction of the blood. Consequently to GS rRT-PCR, the virus isolation technique was performed on the two serum samples with the lowest Ct values.

**Table 2 viruses-14-01545-t002:** Serotype-specific rRT-PCR primers and probes targeting VP2 AHSV genes specific for each serotype.

AHSV	Primers (F/R) and Probe	Sequence 5′-3′
Serotype 1	AHS-1F	GCAAGCGCTGGCACTTG
	AHS-1R	TTCGAACTCATTCCTTACATCAACA
	AHS1P	FAM-AATGTCTTAGATCGTCAACT-MGB
Serotype 2	AHS-2F	CGGAAACTYTGTATTGCCAAA
	AHS-2R	TTGTCRTCCTGATCAACCCTAA
	AHS-2P	Cy5-TGAAGGTGCTTACCCGATCTTTCCACA-BBQ
Serotype 3	AHS-3F	AATTATTACAGCGGAGAATGCAGTT
	AHS-3R	GGTTATGAGTGGGGTGCGA
	AHS-3P	FAM-AGAGTTGAGGTTGCGGGA-MGB
Serotype 4	AHS-4F	TGAGGTGGAACACGAYATGTC
	AHS-4R	GATATGCCCCCTCACAYCTGA
	AHS-4P	VIC-TATCGGRATTTATGTACAATGAG-MGB
Serotype 5	AHS-5F	GAAGAGACAGGCGATTCAAATGA
	AHS-5R	AAAGCCACCCTTTTTGGTACAAA
	AHS-5P	NED -TGTTGARATGCTGAGGC-MGB
Serotype 6	AHS-6F	AGCCAGGGCTTCTTTGCA
	AHS-6R	CTCATGTTCAACCCACTGTACATTAA
	AHS-6P	VIC-GTCATCACCGTAAGCG-MGB
Serotype 7	AHS-7F	AGCCAGGGCTTCTTTGCA
	AHS-7R	CTCATGTTCAACCCACTGTACATTAA
	AHS-7P	VIC-GTCATCACCGTAAGCG-MGB
Serotype 8	AHS-8F	GAAATTATCAGCGGACTGACTAAGAA
	AHS-8R	AAACATCTACCTTTTGCGAATCTTG
	AHS-8P	NED-ACGTGATTCTTTTCCC-MGB
Serotype 9	AHS-9F	TACTGTGTCGGTGAGGGATTTT
	AHS-9R	GCCACGACCGGATATGA
	AHS-9P	FAM-AAACAAACGAAATGTGAA-MGB

**Table 3 viruses-14-01545-t003:** Comparative AHSV detection in EDTA blood samples of unvaccinated horses (#1n, #2n, and #3n).

Sampling Date	Isolation (Serotype) ^†^			GS-rRT-PCR (Ct)		
	#1n	#2n	#3n	#1n	#2n	#3n
0 dpi	Nd	Nd	Nd	Neg	Neg	Neg
1 dpi	Nd	Nd	--	Neg	Neg	--
2 dpi	Nd	Nd	--	Neg	Neg	--
3 dpi	Nd	Pos (S-9)	Nd	Neg	Inc (37.2)	Neg
4 dpi	Pos (S-9)	Pos (S-9)	--	Pos (32.3)	Pos (31.9)	--
5 dpi	Pos (S-9)	Pos (S-9)	--	Pos (28.4)	Pos (27.8)	--
6 dpi	Pos (S-9)	Pos (S-9)	--	Pos (25.8)	Pos (25.9)	--
7 dpi	Pos (S-9)	Pos (S-9)	Pos (S-9)	Pos (22.5)	Pos (22.5)	Pos * (26) (S-9)
8 dpi	Pos (S-9)	Pos (S-9)	--	Pos (18.8)	Pos (19.7)	--
9 dpi	Pos (S-9)	Pos (S-9)	Pos (S-9)	Pos (19.1)	Pos (17.8)	Pos * (23.5) (S-9)
10dpi	--	Pos (S-9)	Pos (S-9)	--	Pos (18.9)	Pos * (21.5) (S-9)

Nd: not done; Inc: inconclusive; Pos: Positive; Neg: Negative. (^†^) Confirmed by GS-rRT-PCR and typed by TS-rRT-PCR. (*) TS rRT-PCR for the nine serotypes were performed, only Pos or Inc. results are reported.

**Table 4 viruses-14-01545-t004:** Comparative AHSV detection in EDTA blood samples of vaccinated horses (#4v and #5v).

Sampling Date	Isolation (Serotype) ^†^		GS-rRT-PCR (Ct)		TS-rRT-PCR * (Ct)	
	#4v	#5v	#4v	#5v	#4v	#5v
0 dpv	Nd	Nd	Neg	Neg	Nd	Nd
3 dpv	Nd	Nd	Neg	Neg	Nd	Nd
6 dpv	Nd	Nd	Neg	Neg	Nd	Nd
9 dpv	Nd	Nd	Neg	Neg	Nd	Nd
13 dpv	Neg	Neg	Pos (34.0)	Neg	Pos S-8 (31.6)	Neg
16 dpv	Pos (S-8)	Neg	Pos (31.5)	Neg	Inc S-6 (35.5) Pos S-8 (28.8)	Neg
21 dpv	Neg	Neg	Pos (30.7)	Pos (36.5)	Inc S-6 (36.1) Pos S-8 (28.2)	Neg
24 dpv	Neg	Neg	Pos (32.2)	Pos (31.4)	Inc S-6 (38.2) Pos S-8 (28.8)	Pos S-2 (32.7)
27 dpv	Neg	Neg	Pos (33.5)	Pos (33.0)	Pos S-8 (31.3)	Pos S-2 (34.7)
30 dpv	Neg	Neg	Pos (34.5)	Pos (33.5)	Pos S-8 (32.6)	Inc S-2 (35.1)
34 dpv/0 dpi	Neg	Neg	Pos (33.3)	Pos (33.1)	Pos S-2 (33.6)	Inc S-2 (36.6)
37 dpv / 3 dpi	Pos (S-2)	Neg	Pos (32.1)	Pos (32.0)	Inc S-2 (38)	Pos S-2 (34)
41 dpv/7 dpi	Neg	Neg	Pos (32.1)	Pos (33.4)	Pos S-2 (32.9)	Pos S-2 (34.4)
43 dpv/9 dpi	Neg	Neg	Pos (32.4)	Pos (33.5)	Pos S-2 (33.2)	Pos S-2 (35)
45 dpv/11 dpi	Neg	Neg	Pos (32.2)	Pos (33.0)	Pos S-2 (34.9)	Inc S-2 (35.5)
48 dpv/14 dpi	Neg	--	Pos (35.3)	--	Pos S-2 (35.0)	--
50 dpv/16 dpi	--	Pos (S-9)	--	Pos (29.6)	--	Pos S-9 (31.4)

Nd: not done; Inc: inconclusive; Pos: Positive; Neg: Negative. (^†^) Confirmed by GS-rRT-PCR and typed by TS-rRT-PCR. * TS rRT-PCR for the nine serotypes were performed, only Pos or Inc. results are reported.

**Table 5 viruses-14-01545-t005:** Comparative AHSV detection in organs of unvaccinated horses (#1n, #2n, and #3n).

	Isolation (Serotype) ^†^			ELISA for Antigen Detection			GS-rRT-PCR (Ct)		
	#1n	#2n	#3n	#1n	#2n	#3n	#1n	#2n	#3n
Heart	Pos (S-9)	Neg	Neg	Pos	Pos	Pos	Pos (21)	Pos (22.3)	Pos (22.5)
Lung	Pos (S-9)	Pos (S-9)	Pos (S-9)	Pos	Pos	Pos	Pos (18.6)	Pos (19.6)	Pos (22.8)
Mediastinal lymph node	Neg	Pos (S-9)	Pos (S-9)	Pos	Pos	Pos	Pos (25.5)	Pos (25.3)	Pos (29.4)
Liver	Neg	Pos (S-9)	Pos (S-9)	Pos	Pos	Pos	Pos (22.4)	Pos (21.3)	Pos (22.1)
Spleen	Pos (S-9)	Pos (S-9)	Pos (S-9)	Pos	Pos	Pos	Pos (21.8)	Pos (18.2)	Pos (19.9)
Kidney	Neg	Pos (S-9)	Pos (S-9)	Pos	Pos	Pos	Pos (24.6)	Pos (24.2)	Pos (28.3)
Mesenteric lymph node	Neg	Pos (S-9)	Pos (S-9)	Neg	Pos	Pos	Nd	Pos (23.6)	Pos (28.4)

Nd: not done; Inc: inconclusive; Pos: Positive; Neg: Negative. (^†^) Confirmed by GS-rRT-PCR and typed by TS-rRT-PCR.

**Table 6 viruses-14-01545-t006:** Comparative AHSV detection in organs of vaccinated horses (#4v and #5v).

	Isolation (Serotype) ^†^		ELISA for Antigen Detection		GS-rRT-PCR (Ct)		TS-rRT-PCR (Ct)	
	#4v	#5v	#4v	#5v	#4v	#5v	#4v	#5v
Heart	Neg	Neg	Nd	Nd	Inc (37.5)	Pos (31.7)	Neg	Inc S-9 (35.9)
Lung	Neg	Neg	Nd	Nd	Pos (35)	Pos (28.2)	Neg	Pos S-9 (31.8)
Mediastinal lymph node	Neg	Neg	Nd	Nd	Inc (37.4)	Pos (31.8)	Neg	Pos S-2 (33.4) Pos S-9 (33.1)
Liver	Neg	Neg	Nd	Nd	Pos (35.2)	Pos (29.7)	Pos S-2 (35) Pos S-9 (34.8)	Pos S-9 (30.4)
Spleen	Neg	Pos (S-9)	Neg	Neg	Pos (30.6)	Pos (25.1)	Pos S-2 (30.8) Pos S-8 (33.7)	Pos S-2 (29) Pos S-9 (28.4)
Kidney	Neg	Neg	Nd	Nd	Neg	Inc (36.5)	Nd	Pos S-9 (34.1)
Mesenteric lymph node	Neg	Neg	Nd	Nd	Neg	Pos (32.9)	Nd	Inc S-2 (39.9) Pos S-9 (32.3)

Nd: not done; Inc: inconclusive; Pos: Positive; Neg: Negative. (^†^) Confirmed by GS-rRT-PCR and typed by TS-rRT-PCR.

**Table 7 viruses-14-01545-t007:** Detection of serotype-specific antibodies (sero-neutralization test) in the vaccinated horses.

			**Horse #4v**								
			**Titre ^†^ to Serotypes Included in Comb2**				**Titre ^†^ to Serotypes Included in Comb1**			**Titre ^†^ to Serotypes Not Included**	
Day pv-C2	Day Pv-C1	Day pi	S-2	S-6	S-7	S-8	S-1	S-3	S-4	S-5	S-9
21	0		<5	<5	<5	<5	<5	<5	<5	<5	<5
24	3		<5	7.5	<5	7.5	<5	<5	<5	<5	<5
27	6		<5	15	<5	7.5	<5	<5	<5	<5	<5
30	9		<5	5	<5	7.5	<5	<5	5	<5	<5
34	13	0	<5	7.5	<5	15	<5	<5	<5	<5	<5
37	16	3	<5	5	<5	10	<5	<5	<5	<5	<5
41	20	7	<5	5	<5	15	<5	<5	<5	<5	<5
43	22	9	<5	10	<5	15	<5	<5	7.5	7.5	<5
45	24	11	7.5	30	<5	15	<5	<5	5	5	<5
48	27	14	30	20	<5	15	7.5	<5	10	7.5	<5
			**Horse #5v**								
			**Titre ^†^ to Serotypes included in Comb2**				**Titre ^†^ to Serotypes included in Comb1**			**Titre ^†^ to Serotypes not included**	
Day pv-C2	Day Pv-C1	Day pi	S-2	S-6	S-7	S-8	S-1	S-3	S-4	S-5	S-9
*21*	*0*		Nd	Nd	Nd	Nd	Nd	Nd	Nd	Nd	Nd
*24*	*3*		Nd	Nd	Nd	Nd	Nd	Nd	Nd	Nd	Nd
*27*	*6*		Nd	Nd	Nd	Nd	Nd	Nd	Nd	Nd	Nd
*30*	*9*		<5	<5	<5	<5	<5	<5	<5	<5	<5
*34*	*13*	*0*	<5	<5	<5	<5	<5	<5	<5	<5	<5
*37*	*16*	*3*	<5	<5	<5	<5	<5	<5	<5	<5	<5
*41*	*20*	*7*	<5	<5	<5	<5	<5	<5	<5	<5	<5
*43*	*22*	*9*	7.5	<5	<5	<5	<5	<5	<5	<5	<5
*45*	*24*	*11*	15	<5	<5	<5	<5	<5	<5	<5	<5
*50*	*29*	*16*	15	<5	<5	<5	<5	<5	<5	<5	<5

Nd: not done. Day pv-C2: day post-vaccination with combination 2. Day pv-C1: day post-vaccination with combination 1. Day pi: day post-infection with AHSV serotype 9 (challenge strain). (^†^) Titre expressed as inverse of dilution.

## Data Availability

The data that support the findings of this study are available from the corresponding author upon reasonable request.

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
