# Peer review of "Clinical, Virological and Immunological Responses after Experimental Infection with African Horse Sickness Virus Serotype 9 in Immunologically Naïve and Vaccinated Horses"

_viruses, 2022, doi:10.3390/v14071545_

Round 1
Reviewer 1 Report
In the manuscript “Clinical, virological and immunological responses after experimental infection with African horse sickness virus serotype 9 in immunologically naïve and vaccinated horses” the authors set out to generate data from controlled AHSV infections in vaccinated and unvaccinated horses. If I measure their outputs (as detailed in the Results and Discussion sections) against their own objectives (listed as a) to c) in the last paragraph of the Introduction) I do not think that in the current version this paper would qualify for meeting its own objectives. I am fully cognisant of the ethical, financial and multiple other constraints on experiments involving horses. Especially in this light, the experiments seem to not have been sufficiently carefully planned and aligned, or optimally executed, to gain the maximum potential benefits that could have been derived from the data. In Experiment 1 two naïve horses were infected with a pathogenic strain of AHSV-9, and in Experiment 2 two horses were vaccinated with the licensed live-attenuated vaccine from South Africa in a non-prescribed protocol, and challenged with AHSV-9 at a time point prior to when full vaccine efficacy is anticipated. In parallel with the challenge, a naïve horses was included as a control in Experiment 2 (biological replicate of Exp 1). However sampling for this horse was not necessarily done on the same days, or for the same set of laboratory investigation, as those in Experiment 1.
The data are theoretically presented chronologically, first from Experiments 1 and then Experiment 2; however there are inconsistencies at more than one position. For example, horse #3n is mentioned in 3.1.2.5 (Exp 1); Table 3 (Exp 1) contains data for horses #3, #4v, #5v; Fig 2 (Exp 2) contains data for horses #1, and #2n and duplicates some information already provided in Fig 3 just in a different format. I suggest you relook the full organisation of the results, and attempt to improve the flow and logic and ease of interpretation. It might work better to present data from both trials simultaneously, sorted per type of experiment? Combine results from the 2 trials into a single figure where appropriate, avoid duplication of information or showing non-informative data, be selective in what is incorporated into figures versus tables versus supplementary materials, improve formatting and quality of both figures and tables.
The discussion is lengthy, but does not clearly convey the message in terms of (1) what novel insights were gained from these experiments, (2) why this work contributes towards your goals of standardisation of AHSV diagnostic testing.
I attach the pdf document of the paper, where I have highlighted grammatical and typographical errors or inconsistencies, and added some comments or suggestions.

Reviewer 2 Report
This article describes the results of two experimental AHSV9 infection studies to characterize the clinical, immunological and virological features, to generate diagnostic reference material and to obtain diagnostic and bioassay data. A major aim of this work was to establish a standardized model for experimental AHSV infections. Two naïve horses were challenged with a virulent AHSV9 strain in the first experiment. One naïve horse and two horses that were vaccinated with the AHSV polyvalent live attenuated vaccine (LAV) were challenged with a virulent AHSV9 strain in the second experiment.
Overall, mostly similar results were observed in the two naïve horses used in experiment 1 and the naïve horse used in experiment 2. All of the naïve horses were successfully infected with the virulent AHSV9, which was detected and isolated from blood and various organs. Both the clinical signs and the necropsy results showed a correlation with the cardiac form of AHS. AHSV was detected early in the naïve horses with RT-PCR and virus isolation in the blood at 3-4 days post-infection. The b-ELISA did not detect serogroup specific VP7 antibodies in two horses, whereas low-level antibodies were observed in one horse at 9-10 days post-infection. However, it should be taken into consideration that it is not expected to detect serogroup specific VP7 antibodies early during the infection because the germinal center-FO B cell high-affinity and isotype switched antibody responses require about 7 days or longer to develop during a primary immune response. AHS is one of the most lethal diseases of horses, and the results of the three naïve horses infected with the virulent AHSV9 served as a pilot study that demonstrated the importance of using various assays for the detection of AHSV. These results may contribute to the generation of diagnostic reference material as well as the improvement and/or optimization of assays for early detection of AHSV in infected horses.
The study design and the interpretation of results of experiment 2 are problematic and inadequate. The vaccination results of experiment 2 should be excluded or sufficiently motivated before this part of the work can be published. Major inadequacies include the small number of LAV vaccinated animals, the lack of a vaccinated-only control horse and the early challenge with the virulent AHSV9 stain without confirming that the attenuated AHSV strains were cleared. Additionally, it is not clear why the vaccinated horses were euthanized at the chosen time points when the results indicated that primary adaptive immune responses against several attenuated AHSV strains as well as the virulent AHSV9 were still taking place.
Specific comments:
Section 2.1. Study design, Lines 111-113: States ‘challenged together with a naïve horse (in a different box) thirteen days after completion of the vaccination protocol (day 34 post vaccination with Comb2/day 13 post vaccination with Comb1)’.
It should be taken into consideration that during the immunization with an attenuated virus vaccine, a primary immune response [innate and adaptive (cellular and humoral)] is induced that results in the clearance of the attenuated virus and the generation of long-term immunological memory. The attenuated virus must first be cleared before the host is challenged with a virulent virus.
The two LAV vaccinated horses were challenged with a virulent AHSV9 stain 13 days post-vaccination with Comb1 (Figure 1), which was too early. It is evident from the results that the attenuated AHSV strains were not cleared, but that the primary adaptive immune responses were still in the process of responding to attenuated AHSV strains and therefore long-term immunological memory was not generated. LAV responses of horse 4: Based on the detection of AHSV RNA and serotype-specific antibodies (Tables 4 and 5), primary adaptive immune responses were still ongoing in response to attenuated AHSV8, AHSV2 and AHSV6, and potentially in the early stages against AHSV1 and AHSV4 (antibodies, but no RNA detected yet). LAV responses of horse 5: Based on the detection of AHSV RNA and serotype-specific antibodies (Tables 4 and 5), primary adaptive immune response was ongoing in response to attenuated AHSV2 only. It is well documented in studies using outbred animals that different immune responses are observed against a vaccine (strong, weak and no responses) in the population.
These horses were challenged with a virulent AHSV9 strain during still ongoing primary adaptive immune responses against attenuated AHSV strains. The adverse effects of this is unknown since LAV vaccinated horse 4 was euthanized 14 days post challenge and LAV vaccinated horse 5 was euthanized 16 days post challenge. Although AHSV9 was only detected in the liver, this was not a memory immune response and it is not known if the primary immune response antibody responses against AHSV6 were effective in providing cross-protection against virulent AHSV9 (horse 4). AHSV6 was never detected in horse 5; it is therefore unlikely that antigen-specific cellular or humoral adaptive immune responses were induced against this strain. This was further indicated by the detection of virulent AHSV9 in the blood and multiple organs. It should also be taken into consideration that the trained innate immune responses likely contributed to the perception that the horses were clinically protected before death, in particular for horse 5. However, the horses were euthanized in the middle of ongoing primary adaptive immune responses, and it remains unknown if this could have resulted in a severe disease or survival.
The LAV vaccinated horses should have been challenged with the virulent AHSV9 at a much later stage (possibly months) as recommended by Coetzer& Guthrie, 2004, referred to in this manuscript. Coetzer& Guthrie, 2004: ‘In South Africa, annual immunization with a live polyvalent attenuated vaccine in late winter or early summer (September to November), which is some time before the peak AHS season (March and April), is advocated and allows immunized animals to respond adequately to the vaccine before possibly being challenged by natural exposure.’
Section 2.2. Horses, Line 130: Data should be included to show that all the horses tested negative for AHSV before the start of the experiments. If they were not tested, provide a reason why testing was not done or necessary. The sample size of the LAV vaccinated horses (2 horses) was too small to be of real experimental statistical significance.
Section 2.4. Vaccination. Line 150: The study did not include a LAV vaccinated-only horse for control in experiment 2. The clinical signs, necropsy and assay results of experiment 2 should have included results from a LAV vaccinated-only horse.
Section 2.7.4. ELISA for Antigen detection, Line 238: ‘values were considered doubtful’. Do not use ‘doubtful’ in this article or any scientific writing. Explain what these intermediate values are (e.g. positive but not significant) and rename here and elsewhere.
Section 3.2.1. Clinical signs and pathology, Line 328: States that ‘previously vaccinated horses did not develop any clinical signs after challenge’. However, only the morning clinical monitoring results were shown for experiment 2 (Figure 2) even though clinical signs were more prominent in the afternoon as observed in experiment 1 (Figure 2). It is therefore unknown if there were clinical signs present during some of the afternoons.
Section 3.2.1. Clinical signs and pathology, Lines 331-333: States that ‘the necropsy revealed no disease-specific lesions, except mild congestion of heart, liver, spleen and kidney’. I am not convinced that ‘mild congestion of heart, liver, spleen and kidney’ is reflective of ‘no disease-specific lesions’ without comparison to necropsy results from LAV vaccinated-only horses at similar time points.
Section 4. Discussion, all the sections about LAV vaccinated horses from experiment 2. Rewrite depending on changes made or exclude if this work is not going to be included in the current publication.
Section 4. Discussion, Lines 462-461: States ‘it is expected this protection would be less efficient if vaccinated animals were infected several months after vaccination’. This statement is completely incorrect. Vaccines are used to generate long-term immunological memory that in turn will protect the host from a virulent pathogen. For the LAV, several courses (boosters) of vaccination may be required to achieve complete immunity against all the AHSV serotypes. See the AHSV review papers used in this manuscript and additional vaccine and immune response review papers for information.
Round 2
Reviewer 1 Report
The manuscript “Clinical, virological and immunological responses after experimental infection with African horse sickness virus serotype 9 in immunologically naïve and vaccinated horses” has now been revised substantially. The data has been reorganised, and new figures prepared. This improves the overall coherence and flow. The reworked discussion also improves clarity. I appreciate that the limitations in study design and sampling protocols cannot be addressed in retrospect. The one aspects that remains insufficiently critiqued by yourself, is the fact that both your challenge of the two vaccinated horses with AHSV-9, and the time of euthanasia of these horses, was too early. The adaptive immune response would not have been fully established at that time, and innate immune processes were still at play. Based on this you should not be making inferences about protection from vaccine-induced immunity, and you should clearly state that this will impact on the detection of viral RNA and antibodies as presented here.
There are still some minor editing issues that can be corrected.
Reviewer 2 Report
Manuscript ID: viruses-1760661
Review report on the revised version of the article entitled ‘Clinical, virological and immunological responses after experimental infection with African horse sickness virus serotype 9 in immunologically naïve and vaccinated horses’
Specific comments:
The authors stated in the response to Reviewer 2: ‘…and it has not been considered essential to include a vaccinated-only animal for the objectives of this study (listed in the last paragraph in the introduction part).’ ‘Moreover, although our main objective is not to evaluate protection, as it was mentioned, the results in our study indicate that in a short time after complete vaccination (13 days) with attenuated multivalent vaccine Ondersteport it is possible to observe protection against the clinical symptoms of the disease if the animals are challenged. We think it is relevant to be reported and discussed, despite design imperfections to assess protection.’
Since the presumed protection was clearly evaluated, it must be included in this article (lines 457-465) that the exclusion of a LAV vaccinated-only horse control was a major shortcoming of the experimental design.
Lines 573-574. States ‘Interestingly, horse #5v did not show neutralizing antibodies to serotype 6 even though the animal was clinically protected against the cross-reactive AHSV serotype 9 challenge strain.’
The horses were euthanized in the middle of ongoing primary adaptive immune responses against the LAV as well as the virulent virus. It is unknown if these horses would have survived or not. Therefore, it is currently assumed by the authors that the horses were clinically protected against the AHSV9 challenge strain. Rewrite here and elsewhere to clarify this (e.g. although it appeared that the horses were clinically protected, additional studies are required to verify that the horses survived).
